# Long-term prognosis after coronary bifurcation PCI—A nationwide observational study

András Katona[1,2☯], Sacharias von Koch[3☯], Pontus Andell[4], Sebastian Völz[5,6], Elmir Omerovic[5,6], Ole Fröbert[1,7,8,9]*, Moman A. Mohammad[3]

1 Faculty of Health, Department of Cardiology, Örebro University, Örebro, Sweden, 2 Albert Szent-Györgyi Clinical Center, Department of Medicine and Cardiology Center, Medical Faculty, University of Szeged, Szeged, Hungary, 3 Department of Cardiology, Clinical Sciences, Lund University, Skåne University Hospital, Lund, Sweden, 4 Department of Physiology and Pharmacology, Karolinska Institutet, and Maine Cardiology, Karolinska University Hospital, Stockholm, Sweden, 5 Department of Cardiology, Sahlgrenska University Hospital, Gothenburg, Sweden, 6 Department of Molecular and Clinical Medicine, Sahlgrenska Academy at University of Gothenburg, Gothenburg, Sweden, 7 Department of Clinical Medicine, Faculty of Health, Aarhus University, Aarhus, Denmark, 8 Department of Clinical Pharmacology, Aarhus University Hospital, Aarhus, Denmark, 9 Steno Diabetes Center Aarhus, Aarhus University Hospital, Aarhus, Denmark

☯ These authors contributed equally.
* ole.frobert@regionorebrolan.se

## Abstract

### Background

Long-term outcomes of percutaneous coronary intervention (PCI) for bifurcation lesions are underexplored. We investigated long-term PCI outcomes for proximal LAD bifurcation lesions involving D1.

### Methods

Using Swedish registries, we included all patients undergoing LAD-D1 bifurcation PCI with drug-eluting stents between 2010 and 2020. Patients were stratified into two groups: simple PCI and complex PCI. The simple PCI group included those with stents in the proximal LAD only, while complex PCI involved the kissing balloon technique or a 2-stent approach for the proximal LAD and D1. A multivariable Cox regression model was used to estimate event rates of major adverse clinical events (MACE), defined as all-cause death or a new myocardial infarction. Secondary outcomes included target segment revascularization or coronary artery by-pass graft surgery (CABG) and definite stent thrombosis.

### Results

A total of 6,796 individuals were analyzed: 2,007 underwent complex PCI and 4,789 simple PCI. Baseline characteristics were comparable between groups. The complex PCI group was slightly younger, more often male, and more frequently taking statins. At 1-year, MACE rates were lower in the complex PCI group (6.2% vs 7.9%; adjusted HR 0.74, 95% CI 0.59-0.93, p=0.010). The result was driven by lower all-cause mortality (3.6% vs. 5.0%;

**Data availability statement:** This study is based on Swedish Coronary Angiography and Angioplasty Registry (SCAAR) data and contains sensitive patient information. Due to patients' privacy and secrecy laws, SCAAR data cannot be publicly available. Upon reasonable request, other researchers can access data through the Uppsala Clinical Research Center, under the provision that the data is accessed onsite. A request can be sent to datauttag@ucr.uu.se. The authors confirm that they had no special data access privileges.

**Funding:** The author(s) received no specific funding for this work.

**Competing interests:** The authors have no conflicts of interest to declare.

adjusted HR 0.73, 95% CI 0.54-0.98, p = 0.036). No significant differences in myocardial infarction, target segment revascularization, CABG, stent thrombosis, stroke, or bleeding were observed between groups, persisting at five years.

## Conclusion

Over a five-year period, complex PCI of LAD/D1 bifurcation lesions was associated with better outcome than simple PCI in a routine clinical setting.

## Introduction

Coronary bifurcation stenting remains a complex and debated area in percutaneous coronary intervention (PCI). Despite advancements in stent technology and pharmacotherapy, bifurcation lesions pose unique challenges, often yielding inferior outcomes compared to non-bifurcation lesions. Single stent strategies, like provisional stenting, are associated with fewer complications and are recommended by some interventionalists. However, certain cases necessitate a double stent strategy to ensure patency in both the main vessel (MV) and side branch (SB).

The effectiveness of different stent techniques, including the double-kissing crush as a two-stent approach, are still under investigation [1]. Further complexity arises from variables such as presence of diabetes mellitus, the choice of P2Y12 inhibitor, the duration of dual antiplatelet therapy, and stent selection [2,3].

Historically, studies have compared simple strategies (MV with optional SB stenting) against more complex ones (planned MV and SB stenting), with the former often showing better clinical outcomes, reduced procedural time, and less resource utilization. In the Nordics, several randomized studies comparing different interventional techniques in coronary bifurcation treatment have been conducted. One study, using third-generation drug-eluting stents, compared MV (and optional SB stenting) with planned MV and SB stenting. At six months, the clinical outcomes after MV stenting did not differ statistically from the more complex strategy of planned stenting of both the MV and the SB, but the MV stenting strategy was associated with reduced procedure and fluoroscopy times and lower rates of procedure-related biomarker elevation [4]. This led to a randomized study comparing MV stenting with and without final kissing balloon dilatation of the MV and SB. The study was neutral concerning major adverse cardiac events [5]. In a later randomized study, clinical outcome after treatment of lesions in large bifurcations by simple or complex stent implantation (preferably the 'culotte technique') also using third generation drug-eluting stents did not differ statistically but in that study the absolute number of events was higher with the simple technique [6].

In a recent randomized study enrolling more than 1,200 patients, PCI with routine optical coherence tomography (OCT) guidance was found to be superior to standard angiography-guided PCI in terms of major adverse cardiac events in revascularization of complex lesions located at a coronary bifurcation. There were no differences between groups in cardiac or all-cause death [7].

Beyond the treatment of left main coronary lesions, the hemodynamic importance of treating coronary side branches is sometimes limited. Additionally, intracoronary imaging techniques are resource-intensive, time-consuming, and require training. Therefore, we found it of interest to investigate the long-term outcomes after PCI of left anterior descending (LAD)-first diagonal branch (D1) bifurcations reflecting regular practice. We chose to focus

on the LAD-D1 bifurcation because it is well-defined anatomically and due to its critical importance in supplying a substantial portion of the myocardium, with the LAD perfusing the anterior wall, septum, and apex, and the D1 perfusing the lateral aspect of the left ventricle. We compared a simple PCI strategy of LAD alone, against a complex PCI strategy involving LAD and D1.

## Materials and methods

All patients undergoing PCI in Sweden are registered in the Swedish Web-system for Enhancement and Development of Evidence-based care in Heart disease Evaluated According to Recommended Therapies (SWEDEHEART) registry, which includes the Swedish Coronary Angiography and Angioplasty Registry (SCAAR) [8]. The registries are sponsored solely by the Swedish health authorities and receive no commercial funding.

In Sweden, a total of 29 hospitals, including nine university hospitals, have facilities for cardiac catheterization. SCAAR records coronary angiographies using approximately 50 variables, and PCI procedures using up to 200 variables. Following each interventional procedure, the physician operator enters relevant clinical characteristics and details of the procedure into the registry. Validation of source data against electronic health records is performed periodically in all hospitals by comparing 50 entered variables in 30 to 40 randomly selected patients per hospital and year with an overall agreement of 95% [8]. SCAAR consistently updates patient vital status data from the national death registry, which maintains a high level of completeness owing to the mandatory use of personal identification numbers. The Swedish Prescribed Drugs Registry was used to collect data on medical therapies dispensed 90 days prior to 14 days after inclusion. Data on pharmacological therapies from the Swedish Prescribed Drugs Registry was linked to SCAAR. The study was approved by the Swedish Ethical Review Authority (Dnr 2023-00201-01), and research was carried out in accordance with appropriate ethical guidelines.

For this study, SCAAR was used to identify all patients in Sweden undergoing LAD-D1 bifurcation PCI using drug-eluting stents between 2010 and 2020. Lesions included were class B1, B2 or C according to the modified American College of Cardiology/American Heart Association lesion morphology classification [9] and classified as bifurcation lesions by the PCI operator. All patients included underwent PCI with stent implantation involving proximal LAD (segment 6). Key exclusion criteria comprised patients with previous PCI and patients with simultaneous stenting of non-LAD coronary arteries. A flowchart illustrating inclusion and exclusion criteria is presented in Fig 1. Patients were divided into two groups according to treatment strategy: simple PCI and complex PCI. The simple PCI group comprised patients undergoing stent implantation in LAD alone. Complex PCI was defined as a procedure involving LAD and D1, including patients undergoing PCI with kissing balloon technique or a 2-stent approach of LAD and D1. A supplementary analysis was also conducted with three groups comparing simple PCI, kissing balloon technique and a 2-stent approach.

The primary outcome variable was major adverse clinical events (MACE), which included all-cause death and nonfatal myocardial infarction, assessed at 1-year follow-up. Secondary outcome variables, evaluated at both 1 and 5 years, included the individual components of MACE, target segment revascularization or coronary artery by-pass graft surgery (CABG), and definite stent thrombosis, with an additional assessment of overall MACE at 5 years. To evaluate the comparability of the two groups regarding potential unmeasured confounders, we also assessed the incidence of stroke and bleeding events at 1 and 5 years. All outcomes were ascertained up to January 15, 2022, with complete follow-up for all patients. Outcome definitions are presented in S1 Table.

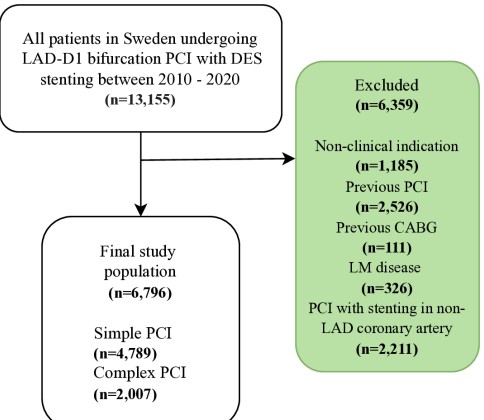

**Fig 1. Flowchart.** Flowchart illustrating eligible patients after inclusion and exclusion criterion. The final study population consisted of 6,796 patients. CABG = coronary artery by-pass graft surgery; D1 = first diagonal branch; DES = drug-eluting stent; LAD = left anterior descending; LM = left main; PCI = percutaneous coronary intervention.

Continuous data are presented as means with standard deviation and differences in normally distributed continuous variables were assessed using independent t-test. Categorical data were presented as counts and percentages and analyzed using the chi-square test. Outcomes were assessed using unadjusted Kaplan-Meier estimates along with univariable and multivariable Cox regression models. The multivariable Cox regression model was employed to adjust for both known and potential confounding factors. In this analysis we adjusted for sex, age, inclusion year, diabetes mellitus, hypertension, hyperlipidemia, smoking status (non-smokers, previous smokers, and active smokers), indication (chronic coronary syndrome, unstable angina, NSTEMI, and STEMI), lesion classification (B1, B2, and C) and use of intracoronary imaging. A subgroup analysis was carried out on 1-year MACE using the multivariable Cox regression model and interaction p-values. Four predefined subgroups were analyzed: sex, age (<80 years vs ≥ 80 years), diabetes mellitus and use of intracoronary imaging. Results from all Cox regression analyses are presented with hazard ratio (HR) and 95% confidence interval (CI). All analyses were conducted on complete case data, and the proportion of missing values in the variables of interest were low (Table 1).
We applied the E-value methodology in a sensitivity analysis to determine the influence of unmeasured confounding [10]. This technique calculates the smallest association strength that an unmeasured confounder would need to negate an observed statistically significant effect, thus addressing potential residual confounding.

A two-sided p-value less than 0.05 was considered statistically significant. Data management and all statistical analyses were done in STATA SE (version 17.0; StataCorp, Texas, LLC).

## Results

A total of 6,796 patients were eligible for the analysis (Fig 1), with 2,007 in the complex PCI group and 4,789 in the simple PCI group. Baseline variables were generally well balanced between the two groups (Table 1). Individuals in the complex PCI group were slightly younger (66.6 ± 11.1 vs 67.3 ± 11.3 years, p = 0.013), more often male (77.6% vs 74.9%, p = 0.015), and more likely to receive statin therapy (93.6% vs 91.1%, p < 0.001). As expected, contrast use was significantly higher in the complex PCI group (194.3 ± 76.9 mL vs 153.5 ± 62.4 mL, p < 0.001) and there were also subtle differences in bifurcation lesion classifications.

**Table 1. Baseline characteristics.** ACEi = angiotensin-converting-enzyme inhibitor; ARB = angiotensin II receptor blocker; ASA = acetylsalicylic acid; CAD = coronary artery disease; NSTEMI = non-ST-elevation myocardial infarction; PCI = percutaneous coronary intervention; STEMI = ST-elevation myocardial infarction.

| | | Simple PCI (n = 4789) | Complex PCI (n = 2007) | p-value | Missing (%) |
|---|---|---|---|---|---|
| **Patient demographics** | | | | | |
| Inclusion time | 2010–2012 | 666 (13.9%) | 291 (14.5%) | 0.007 | 0.0 |
| | 2013–2015 | 1289 (26.9%) | 478 (23.8%) | | |
| | 2016–2018 | 1645 (34.3%) | 672 (33.5%) | | |
| | 2019–2020 | 1189 (24.8%) | 566 (28.2%) | | |
| Age, mean (SD) | | 67.3 (11.3) | 66.6 (11.1) | 0.013 | 0.0 |
| Age ≥ 80 years | | 707 (14.8%) | 238 (11.9%) | 0.002 | 0.0 |
| Sex (male) | | 3585 (74.9%) | 1558 (77.6%) | 0.015 | 0.0 |
| Smoking status | Non-smoker | 2183 (48.1%) | 909 (47.4%) | 0.78 | 5.0 |
| | Previous smoker | 1589 (35.0%) | 688 (35.9%) | | |
| | Active smoker | 770 (17.0%) | 319 (16.6%) | | |
| **Comorbidities** | | | | | |
| Diabetes mellitus | | 825 (17.2%) | 329 (16.4%) | 0.40 | 0.0 |
| Hypertension | | 1159 (24.2%) | 471 (23.5%) | 0.52 | 0.0 |
| Hyperlipidemia | | 1619 (34.1%) | 673 (33.9%) | 0.86 | 0.9 |
| Previous myocardial infarction | | 265 (5.6%) | 113 (5.7%) | 0.87 | 1.4 |
| Previous stroke | | 200 (4.2%) | 79 (3.9%) | 0.65 | 0.0 |
| Renal failure | | 176 (3.7%) | 59 (2.9%) | 0.13 | 0.0 |
| Heart failure | | 113 (2.4%) | 50 (2.5%) | 0.75 | 0.0 |
| **Procedural characteristics** | | | | | |
| Indication | Stable CAD | 908 (19.0%) | 413 (20.6%) | 0.46 | 0.0 |
| | Unstable angina | 1059 (22.1%) | 430 (21.4%) | | |
| | NSTEMI | 1326 (27.7%) | 555 (27.7%) | | |
| | STEMI | 1496 (31.2%) | 609 (30.3%) | | |
| Lesion classification | B1 bifurcation | 1632 (34.1%) | 527 (26.3%) | <0.001 | 0.0 |
| | B2 bifurcation | 2151 (44.9%) | 1006 (50.1%) | | |
| | C bifurcation | 1006 (21.0%) | 474 (23.6%) | | |
| Killip class | 1 | 3910 (94.6%) | 1707 (96.1%) | 0.086 | 13.1 |
| | 2 | 140 (3.4%) | 46 (2.6%) | | |
| | 3 | 34 (0.8%) | 13 (0.7%) | | |
| | 4 | 48 (1.2%) | 11 (0.6%) | | |
| Vascular approach | Radial | 4137 (86.5%) | 1762 (87.9%) | 0.29 | 0.1 |
| | Femoral | 582 (12.2%) | 218 (10.9%) | | |
| | Other | 66 (1.4%) | 25 (1.2%) | | |
| Use of intracoronary imaging | | 303 (6.3%) | 156 (7.8%) | 0.030 | 0.0 |
| Contrast volume, mL, mean (SD) | | 153.5 (62.4) | 194.3 (76.9) | <0.001 | 0.0 |
| **Discharge medication** | | | | | |
| ASA | | 4328 (90.4%) | 1827 (91.0%) | 0.40 | 0.0 |
| P2Y12 antagonists | | 4545 (94.9%) | 1920 (95.7%) | 0.18 | 0.0 |
| Statin | | 4362 (91.1%) | 1878 (93.6%) | <0.001 | 0.0 |
| Beta-blocker | | 3816 (79.7%) | 1562 (77.8%) | 0.086 | 0.0 |
| ACEi/ARB | | 3336 (69.7%) | 1403 (69.9%) | 0.84 | 0.0 |
| Calcium channel blocker | | 984 (20.5%) | 429 (21.4%) | 0.44 | 0.0 |

The primary outcome of MACE at 1-year was statistically significant in the unadjusted analysis, occurring in 124 (6.2%) patients in the complex PCI group versus 378 (7.9%) in the simple PCI group; the adjusted analysis confirmed a HR of 0.74 (95% CI 0.59-0.93, p = 0.010) (Table 2 and Fig 2). Complex PCI was also associated with lower rates of all-cause mortality (3.6% vs 5.0%; adjusted HR 0.73, 95% CI 0.54-0.98, p = 0.036). No difference was observed for myocardial infarction. Target segment revascularization or CABG and definite stent thrombosis numerically favored simple PCI but did not reach statistical significance in the adjusted analysis. There were no differences between groups in stroke or bleeding events.

At five years the pattern persisted. MACE occurred in 283 (17.0%) patients in the complex PCI group and in 820 (19.8%) patients in the simple PCI group (adjusted HR 0.83, 95% CI 0.72-0.96, p = 0.010).

**Table 2. Outcomes. *Adjusted for: age, sex, inclusion year, diabetes mellitus, hypertension, hyperlipidemia, smoking status (non-smokers, previous smokers and active smokers), use of intracoronary imaging, indication (stable coronary artery disease, unstable angina, NSTEMI and STEMI) and lesion classification (B1 bifurcation, B2 bifurcation and C bifurcation).**

| | Events, (KM, %) | | HR, (95% CI) | |
|---|---|---|---|---|
| | Simple PCI (n = 4789) | Complex PCI (n = 2007) | Unadjusted | Adjusted* |
| **1-year** | | | | |
| MACE | 378 (7.9) | 124 (6.2) | 0.77 (0.63-0.95) p-value: 0.014 | 0.74 (0.59-0.93) p-value: 0.010 |
| All-cause mortality | 242 (5.0) | 72 (3.6) | 0.71 (0.54-0.92) p-value: 0.009 | 0.73 (0.54-0.98) p-value: 0.036 |
| Myocardial infarction | 156 (3.3) | 54 (2.7) | 0.82 (0.60-1.11) p-value: 0.202 | 0.73 (0.53-1.03) p-value: 0.070 |
| Target segment revascularization or CABG | 178 (3.8) | 94 (4.8) | 1.26 (0.98-1.61) p-value: 0.075 | 1.16 (0.90-1.51) p-value: 0.251 |
| Definite stent thrombosis | 7 (0.1) | 7 (0.3) | 2.37 (0.83-6.77) p-value: 0.106 | 2.38 (0.76-7.41) p-value: 0.135 |
| Bleeding event | 171 (3.7) | 71 (3.6) | 0.98 (0.74-1.29) p-value: 0.884 | 0.97 (0.72-1.30) p-value: 0.833 |
| Stroke | 41 (0.9) | 19 (1.0) | 1.10 (0.64-1.89) p-value: 0.740 | 1.09 (0.62-1.94) p-value: 0.759 |
| **5-year** | | | | |
| MACE | 820 (19.8) | 283 (17.0) | 0.83 (0.72-0.95) p-value: 0.007 | 0.83 (0.72-0.96) p-value: 0.010 |
| All-cause mortality | 550 (13.6) | 174 (10.8) | 0.76 (0.64-0.90) p-value: 0.002 | 0.77 (0.64-0.93) p-value: 0.007 |
| Myocardial infarction | 338 (9.5) | 134 (9.3) | 0.95 (0.78-1.16) p-value: 0.617 | 0.90 (0.73-1.12) p-value: 0.343 |
| Target segment revascularization or CABG | 311 (7.6) | 150 (8.5) | 1.16 (0.96-1.41) p-value: 0.134 | 1.09 (0.89-1.34) p-value: 0.405 |
| Definite stent thrombosis | 13 (0.3) | 10 (0.6) | 1.85 (0.81-4.21) p-value: 0.145 | 2.00 (0.83-4.85) p-value: 0.124 |
| Bleeding event | 302 (7.7) | 134 (8.2) | 1.06 (0.87-1.30) p-value: 0.565 | 1.11 (0.90-1.38) p-value: 0.321 |
| Stroke | 151 (4.2) | 63 (4.1) | 1.01 (0.75-1.35) p-value: 0.959 | 0.99 (0.73-1.35) p-value: 0.952 |

CI = confidence interval; HR = hazard ratio; KM = Kaplan-Meier estimate; MACE = major adverse clinical events; NSTEMI = non-ST-elevation myocardial infarction; PCI = percutaneous coronary intervention; STEMI = ST-elevation myocardial infarction.

We also calculated hazard ratios for the primary composite outcome measure of 1-year MACE in four predefined subgroups – sex, age (<80 years vs ≥80 years), presence of diabetes, and use of intracoronary imaging. Across all subgroups, the findings were consistent with the primary composite outcome result and no statistically significant interactions were found (S1 Fig).

In a supplementary analysis investigating the event rate of MACE for simple PCI, kissing balloon technique and 2-stent approach, the results were in line with the main analysis showing numerically higher rates of MACE for simple PCI compared to both kissing balloon technique and 2-stent approach (S2 Fig).

S2 Table reports the E -values for MACE and all-cause mortality at 1-year and five years. The e-values for the point estimates ranged from 1.53 to 2.10, indicating that a confounder would need to be associated with both the exposure and the outcome by a HR of 1.53 each, beyond the measured covariates, to explain away the observed association.

## Discussion

In this nationwide, comprehensive, observational study simple PCI of LAD-D1 lesions was associated with a higher risk of MACE compared with complex PCI until five years of follow-up. Our study holds lessons for real-world applicability of findings from randomized trials as well as caveats in the interpretation of observational data.

Earlier randomized trials showed that simple MV stenting technique provides excellent clinical results similar to those of more complex strategies in patients with coronary bifurcation lesions although the absolute number of events in some studies were in favor of complex PCI without reaching statistical significance [4–6]. In more recent randomized trials, the use of intracoronary imaging to guide PCI has shown superiority concerning MACE compared to angiography-guided PCI [11,12] but these trials were not dedicated bifurcation studies. One randomized study on coronary bifurcations showed promising results concerning stent apposition with intracoronary imaging compared with angiography-guided PCI [13], but it was not powered for clinical endpoints. In a trial published 2023 and enrolling 1201 patients with coronary bifurcation lesions allocation to OCT-guided PCI was associated a lower incidence of MACE at 2 years compared with angiography-guided PCI [7].

We found that intracoronary imaging was used in only one out of 15 bifurcation lesions (Table 1), consistent with the fact that trials demonstrating clinical advantage of imaging-guided PCI were mostly published after the observation period of this study ended. It is implausible that the low use of imaging in our study could have influenced our findings of a reduced risk of MACE with complex PCI (see also S1 Fig).

The Kaplan-Meier curves for all-cause death and myocardial infarction (Fig 2) diverged early after PCI, a pattern that was similarly reflected in the event curves for MACE. Even though we adjusted for ten different variables, we cannot exclude that residual confounding— such as frail patients being more likely to receive simple PCI—might partly explain this early difference in events. Another possibility is confounding by indication, where patients with more severe disease more often receive simple PCI, which could also contribute to the early separation of the Kaplan-Meier curves [14]. There are, however, also indications that our findings are not exclusively due to confounding. An early uptick in myocardial infarction occurrence could be explained by stenting over the first diagonal branch, leading to procedure-related infarction, which could theoretically increase the risk of death.

This study has important limitations. The TIMI myocardial perfusion grade could have provided important information that operators likely used in some cases to guide the type of PCI. However, this information is not available in SCAAR. Also, information on Medina bifurcation lesion classification [15], which could have added granularity to the

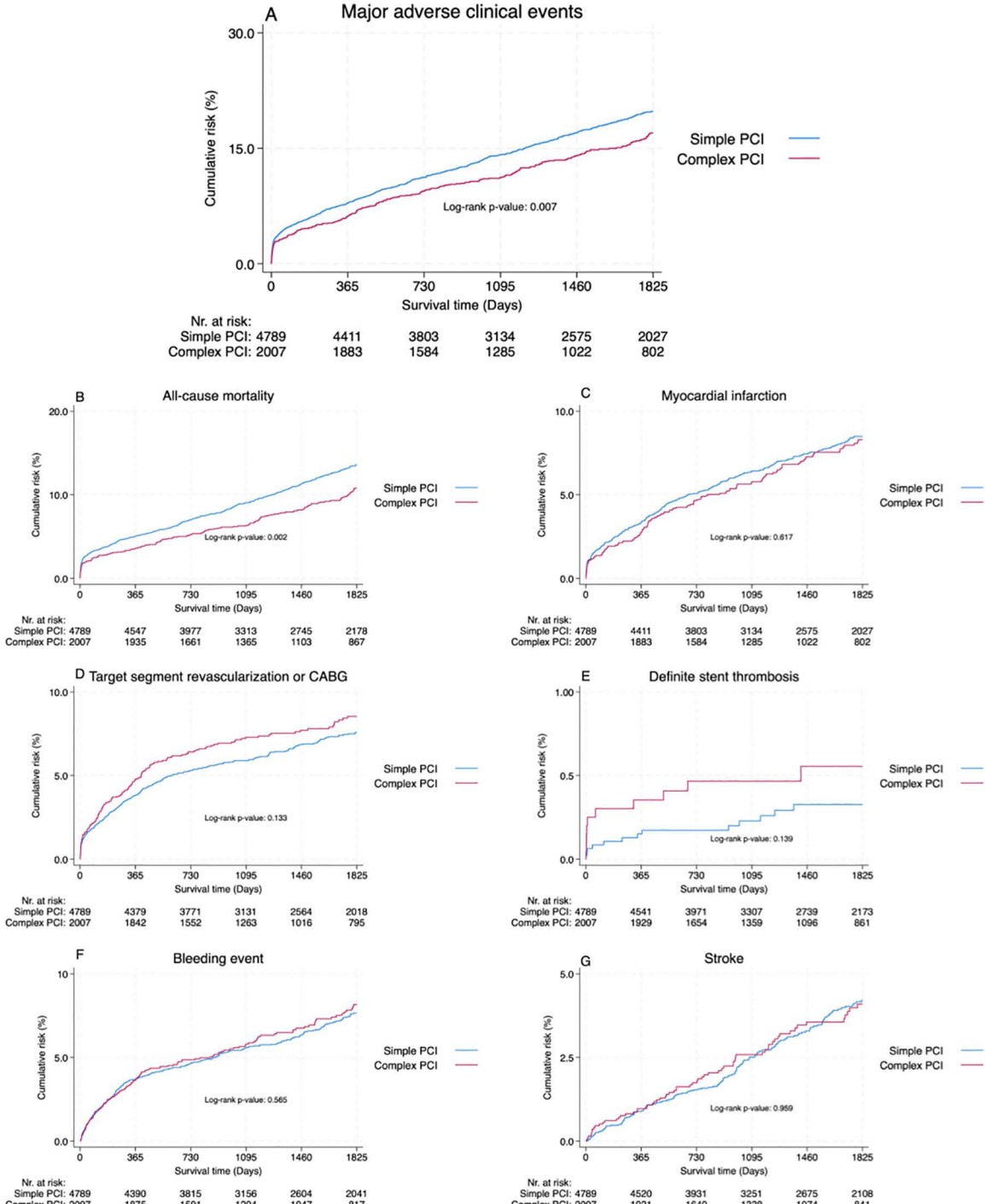

**Fig 2. Outcome.** Kaplan-Meier curves illustrating the 5-year event rate of (A) major adverse clinical events, (B) all-cause mortality, (C) myocardial infarction, (D) target segment revascularization or CABG, (E) definite stent thrombosis, (F) bleeding event, and (G) stroke. Major adverse cardiovascular event was defined as all-cause mortality or myocardial infarction. CABG = coronary artery by-pass graft surgery; PCI = percutaneous coronary intervention.

interpretation of findings, was not available. As lesion morphology assessed by the Medina classification could influence the selection and outcomes of PCI strategies, its absence may affect the interpretation of the comparative effectiveness of simple versus complex PCI approaches. Also, we did not have data on cause-specific mortality. The observational nature of the study provides evidence of association only, not causation. We cannot exclude residual confounding or selection bias. For example, patients in the simple PCI group were slightly older, included more women, and were less frequently prescribed statins. Additionally, this group had a larger sample size, with more patients recruited before 2015, which may reflect temporal changes in practice and outcomes. These differences could have influenced the observed results and should be considered when interpreting the findings. However, the similarity in stroke and bleeding events between the two groups, coupled with E-values indicating moderate to high robustness against unmeasured confounding, supports the reliability of the data. Additionally, the data stems from a single country only, and follow-up ended in 2020, before the publication of seminal randomized studies indicating outcome improvement with imaging-guided PCI.

## Conclusion

Over a ten-year period based on comprehensive Swedish registry data, complex PCI of LAD/D1 bifurcation lesions was associated with better outcome than simple PCI in a routine clinical setting. Our findings are in line with numerical differences in earlier randomized trials not including coronary imaging.

## Supporting information

**S1 Fig. Subgroup analysis.** Forest plot illustrating subgroup analysis. Subgroups were analyzed on major adverse clinical events after 1-year follow up. The subgroups were analyzed using a multivariable Cox regression model including age, sex, inclusion year, diabetes mellitus, hypertension, hyperlipidemia, smoking status (non-smokers, previous smokers, and active smokers), use of intracoronary image, indication (stable coronary artery disease, unstable angina, NSTEMI and STEMI) and lesion classification (B1 bifurcation, B2 bifurcation and C bifurcation). NSTEMI = non-ST-elevation myocardial infarction; PCI = percutaneous coronary intervention; STEMI = ST-elevation myocardial infarction.
(TIF)

**S2 Fig. Kissing balloon technique and two-stent approach.** Kaplan-Meier curves illustrating the 5-year event rate of major adverse clinical events comparing Simple PCI, kissing balloon technique and a two-stent approach. Major adverse cardiovascular event was defined as all-cause mortality or myocardial infarction. D1 = first diagonal branch; LAD = left anterior descending artery; PCI = percutaneous coronary intervention.
(TIF)

**S1 Table. Outcome definition.** CABG = coronary artery by-pass graft surgery; ICD = international classification of diseases; PCI = percutaneous coronary intervention; SCAAR = Swedish Coronary Angiography and Angioplasty registry.
(DOCX)

**S2 Table. E-values.** E-values were calculated for all outcomes with statistically significant differences between Simple PCI and Complex PCI. E-values were calculated using the adjusted and unadjusted effect estimates from Fig 2. Outcome. CI = confidence interval; MACE = major adverse clinical events.
(DOCX)

## Author contributions

**Conceptualization:** András Katona, Sacharias von Koch, Ole Fröbert, Moman A. Mohammad.

**Data curation:** Sacharias von Koch.

**Formal analysis:** Sacharias von Koch.

**Investigation:** Ole Fröbert, Moman A. Mohammad.

**Methodology:** András Katona, Sacharias von Koch, Pontus Andell, Sebastian Völz, Elmir Omerovic, Ole Fröbert, Moman A. Mohammad.

**Supervision:** Ole Fröbert, Moman A. Mohammad.

**Validation:** Elmir Omerovic, Ole Fröbert, Moman A. Mohammad.

**Visualization:** Ole Fröbert.

**Writing – original draft:** András Katona, Sacharias von Koch, Ole Fröbert.

**Writing – review & editing:** András Katona, Sacharias von Koch, Pontus Andell, Sebastian Völz, Elmir Omerovic, Moman A. Mohammad.

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
