## [Decision Letter · Decision Letter 0]

22 Dec 2024

PONE-D-24-45238Long-term prognosis after coronary bifurcation PCI - A nationwide observational studyPLOS ONE

Dear Dr. Fröbert,

Thank you for submitting your manuscript to PLOS ONE. After careful consideration, we feel that it has merit but does not fully meet PLOS ONE’s publication criteria as it currently stands. Therefore, we invite you to submit a revised version of the manuscript that addresses the points raised during the review process. Please submit your revised manuscript by Feb 05 2025 11:59PM. If you will need more time than this to complete your revisions, please reply to this message or contact the journal office at plosone@plos.org . Please include the following items when submitting your revised manuscript:

We look forward to receiving your revised manuscript.

Kind regards,

Ahmed Qasim Mohammed Alhatemi

Academic Editor

PLOS ONE

5. Please include your tables as part of your main manuscript and remove the individual files. Please note that supplementary tables should remain as separate "supporting information" files.

6. We notice that your supplementary figures are uploaded with the file type 'Figure'. Please amend the file type to 'Supporting Information'. Please ensure that each Supporting Information file has a legend listed in the manuscript after the references list.

Reviewers' comments:

Reviewer's Responses to Questions

**Comments to the Author**

1. Is the manuscript technically sound, and do the data support the conclusions?

Reviewer #1: Yes

Reviewer #2: Yes

Reviewer #3: Yes

Reviewer #4: Yes

2. Has the statistical analysis been performed appropriately and rigorously?

Reviewer #1: I Don't Know

Reviewer #2: Yes

Reviewer #3: Yes

Reviewer #4: Yes

3. Have the authors made all data underlying the findings in their manuscript fully available?

Reviewer #1: Yes

Reviewer #2: No

Reviewer #3: Yes

Reviewer #4: Yes

4. Is the manuscript presented in an intelligible fashion and written in standard English?

Reviewer #1: Yes

Reviewer #2: Yes

Reviewer #3: Yes

Reviewer #4: Yes

5. Review Comments to the Author

Reviewer #1: This real world data from a comprehensive registry is very valuable. Although at variance with what randomized clinical trials suggest it is reassuring that when a patient is deemed to need 2 stents or kissing balloon dilation for the LAD/D1 bifurcation, the long term result is good and should not prevent interventional cardiologists from performing this procedure when deemed required to achieve an acceptable result.

Reviewer #2: I would like to congratulate you on conducting such a comprehensive and valuable study that contributes significantly to the field of coronary bifurcation PCI. The use of the SWEDEHEART registry and the long-term follow-up data strengthens the robustness of your findings. However, I would like to kindly point out a few considerations. While you acknowledge the potential impact of selection bias and residual confounding, the absence of Medina bifurcation lesion classification in the dataset may have limited the granularity of your analysis, potentially affecting the comparison between the treatment strategies. Although you briefly discuss this limitation, further consideration of their potential impact could add depth to the interpretation of your findings. I believe your study provides valuable insights and the results remain compelling. I look forward to seeing future studies that may address these factors to further enhance the clinical relevance of the findings.

Reviewer #3: The manuscript provides valuable insights into the long-term outcomes of complex versus simple PCI in the treatment of LAD-D1 bifurcation lesions, grounded in real-world data. Despite its limitations, such as the observational nature and incomplete lesion classification, the study’s findings are significant for clinical practice, suggesting that complex PCI may provide better outcomes in routine care. The authors made it clear that the use of intravascular imaging was deficient, which should not be the case nowadays.

Reviewer #4: The study is exhaustive with large number of patients, and the article is well written with good statistics.

1. Since the study was predominantly performed in LAD D1 bifurcation, the heading can be modified as LAD D1 bifurcation lesions.

2. The stent characteristics is not available in both categories, especially if they are all drug eluting or any bare metal stents were used or not, in both categories. If the information is not available it can be mentioned as a minor limitation in study.

3. In simple stenting, the patients were slightly older, higher incidence of diabetes, lesser intake of statins, increased female patients, more early recruitment (<2015) and comparatively double the number of patients which could have influenced the results. This can be mentioned a minor limitation in discussion.

6. PLOS authors have the option to publish the peer review history of their article (what does this mean? ). If published, this will include your full peer review and any attached files.

**Do you want your identity to be public for this peer review?** For information about this choice, including consent withdrawal, please see our Privacy Policy .

Reviewer #1: **Yes: ** Dr Shukri Merza AlSaif

Reviewer #2: **Yes: ** Pooria Ahmadi

Reviewer #3: **Yes: ** Sultan Alotaibi

Reviewer #4: **Yes: ** Mark Christopher Arokiaraj

---

## [Author Response · Author response to Decision Letter 1]

30 Dec 2024

Editorial Board Comments for the Author.

We thank the editor for their valuable suggestions and helpful comments. The manuscript has been formatted in accordance with the points outlined below. We took the liberty of formatting it before activating track changes and then enabled track changes for subsequent modifications to the manuscript text.

Reply: we have formatted the manuscript in accordance with PLOS ONE's style requirements.

Reply. Unfortunately, Swedish law and legal requirements do not permit immediate data sharing. In line with previous publications in Plos One from our registry (e.g. PMID: 35025950), we have included a statement addressing this issue (P 14):

Data Availability

SWEDEHEART does not allow individual data sharing to third parties. Access to aggregated data might be granted following review by the SWEDEHEART steering committee. Such requests can be submitted to the SWEDEHEART steering committee for consideration. For contact details see https://www.ucr.uu.se/swedeheart/kontakt/styrgrupp.

Reply. The corresponding author (Ole Frobert) has updated this information.

Reply. We have done this in the revised version.

5. Please include your tables as part of your main manuscript and remove the individual files. Please note that supplementary tables should remain as separate "supporting information" files.

Reply. We have done this in the revised version.

6. We notice that your supplementary figures are uploaded with the file type 'Figure'. Please amend the file type to 'Supporting Information'. Please ensure that each Supporting Information file has a legend listed in the manuscript after the references list.

Reply. We have changed this in the revised version and amended file types as requested.

Reply. We have reviewed the reference list and not made any changes to the reference list in the revised version.

Review Comments to the Author

Reviewer #1: This real world data from a comprehensive registry is very valuable. Although at variance with what randomized clinical trials suggest it is reassuring that when a patient is deemed to need 2 stents or kissing balloon dilation for the LAD/D1 bifurcation, the long term result is good and should not prevent interventional cardiologists from performing this procedure when deemed required to achieve an acceptable result.

We thank the reviewer for this kind feedback.

Reviewer #2: I would like to congratulate you on conducting such a comprehensive and valuable study that contributes significantly to the field of coronary bifurcation PCI. The use of the SWEDEHEART registry and the long-term follow-up data strengthens the robustness of your findings. However, I would like to kindly point out a few considerations. While you acknowledge the potential impact of selection bias and residual confounding, the absence of Medina bifurcation lesion classification in the dataset may have limited the granularity of your analysis, potentially affecting the comparison between the treatment strategies. Although you briefly discuss this limitation, further consideration of their potential impact could add depth to the interpretation of your findings. I believe your study provides valuable insights and the results remain compelling. I look forward to seeing future studies that may address these factors to further enhance the clinical relevance of the findings.

We thank the reviewer for valuable suggestions and helpful comments. In the Discussion under limitations we have added (P13, L305):

As lesion morphology assessed by the Medina classification could influence the selection and outcomes of PCI strategies, its absence may affect the interpretation of the comparative effectiveness of simple versus complex PCI approaches.

Reviewer #3: The manuscript provides valuable insights into the long-term outcomes of complex versus simple PCI in the treatment of LAD-D1 bifurcation lesions, grounded in real-world data. Despite its limitations, such as the observational nature and incomplete lesion classification, the study’s findings are significant for clinical practice, suggesting that complex PCI may provide better outcomes in routine care. The authors made it clear that the use of intravascular imaging was deficient, which should not be the case nowadays.

We thank the reviewer for this kind feedback.

Reviewer #4: The study is exhaustive with large number of patients, and the article is well written with good statistics.

We thank the reviewer for valuable suggestions and helpful comments.

1. Since the study was predominantly performed in LAD D1 bifurcation, the heading can be modified as LAD D1 bifurcation lesions.

Reply. We are not completely opposed to changing the title, and if the editors favor a change, we will comply. However, we prefer the current title because it highlights LAD/D1 bifurcations as a case in point to illustrate the potential maximal impact of different stenting techniques. We aimed to make this clear in the Introduction (P4, L110-13):

We chose to focus on the LAD-D1 bifurcation because it is well-defined anatomically and due to its critical importance in supplying a substantial portion of the myocardium, with the LAD perfusing the anterior wall, septum, and apex, and the D1 perfusing the lateral aspect of the left ventricle.

2. The stent characteristics is not available in both categories, especially if they are all drug eluting or any bare metal stents were used or not, in both categories. If the information is not available it can be mentioned as a minor limitation in study.

Thanks. Only drug-eluting stents were used. This is mentioned in the abstract (the word ‘stent’ was missing, now corrected) and methods (P5, L139).

3. In simple stenting, the patients were slightly older, higher incidence of diabetes, lesser intake of statins, increased female patients, more early recruitment (<2015) and comparatively double the number of patients which could have influenced the results. This can be mentioned a minor limitation in discussion.

Reply. Thanks. In the revised version we have mentioned these differences (except diabetes, which did not differ between groups) under limitations (P13, L310-15):

For example, patients in the simple PCI group were slightly older, included more women, and were less frequently prescribed statins. Additionally, this group had a larger sample size, with more patients recruited before 2015, which may reflect temporal changes in practice and outcomes. These differences could have influenced the observed results and should be considered when interpreting the findings.

---

## [Editor Report · Decision Letter 1]

2 Jan 2025

Long-term prognosis after coronary bifurcation PCI - A nationwide observational study

PONE-D-24-45238R1

Dear Dr. Fröbert,

We’re pleased to inform you that your manuscript has been judged scientifically suitable for publication and will be formally accepted for publication once it meets all outstanding technical requirements.

Kind regards,

Ahmed Qasim Mohammed Alhatemi

Academic Editor

PLOS ONE
---

## [Editor Report · Acceptance letter]

PONE-D-24-45238R1

PLOS ONE

Dear Dr. Fröbert,

I'm pleased to inform you that your manuscript has been deemed suitable for publication in PLOS ONE. Congratulations! Your manuscript is now being handed over to our production team.

Kind regards,

on behalf of

Dr. Ahmed Qasim Mohammed Alhatemi

Academic Editor

PLOS ONE